# Influence of vitamin E and carcass feeding supplementation on fecal glucocorticoid and androgen metabolites in male black-footed ferrets (*Mustela nigripes*)

Rachel M. Santymire[1]*, Shana R. Lavin[1¤a], Heather Branvold-Faber[2¤b],
Julie Kreeger[2¤c], Judy Che-Castaldo[1], Michelle Rafacz[1¤d], Paul Marinari[2,3]

1 Conservation and Science Department, Lincoln Park Zoo, Chicago, Illinois, United States of America,
2 United States Fish and Wildlife Service, National Black Footed Ferret Conservation Center, Carr, Colorado,
United States of America, 3 Smithsonian Conservation Biology Institute, Front Royal, Virginia, United States
of America

¤a Current address: Animals, Science, and Environment, Disney's Animal Kingdom, Orlando, Florida, United
States of America.
¤b Current address: Southside Animal Hospital, Anchorage, Alaska, United States of America.
¤c Current address: Bovey, Minnesota, United States of America.
¤d Current address: Columbia College, Chicago, Illinois, United States of America.
* rsantymire@lpzoo.org

Cellular and Molecular Biology, INDIA

**Data Availability Statement:** All relevant data are
within the manuscript and its Supporting
Information files.

## Abstract

In recent years, the *ex situ* population of the endangered black-footed ferret (*Mustela nigripes*; ferret) has experienced a decline in normal sperm morphology (from 50% to 20%), which may be linked to inbreeding depression and/or a dietary change. We examined the effects of adding carcass and vitamin E to the diet on stress and reproductive biomarkers in male ferrets (*n* = 42 males including 16 juveniles and 26 adults) housed at the U.S. Fish and Wildlife National Black-footed Ferret Conservation Center (Carr, CO, USA). Fecal samples (3x/week) were collected from November and December (pre-breeding season, no diet change), February through May (breeding season, diet change) and June (post-breeding season, diet change) and analyzed for fecal glucocorticoid metabolites (FGM) via a cortico-sterone enzyme immunoassay (EIA). A subset of samples from adult males (*n* = 15) were analyzed for fecal androgen metabolites (FAM) via a testosterone EIA. We first used a linear mixed effects model to identify the important fixed effects among meat treatment, vitamin E treatment, age class (juvenile or adult), and all possible interactions on each hormone. We then examined the important factor's effects across seasons using the non-parametric Friedman test. We found that age did not influence (p = 0.33) FGMs; however there was a significant effect of meat treatment on FGM (p = 0.04) and an effect of vitamin E on FAMs (p<0.10). When fed carcass, FGMs declined (p<0.001) from pre- to the during the breeding season time period, but was similar (p>0.05) between during and post-breeding season periods. Males that were not fed carcass had higher (p<0.05) FGMs during the breeding season compared to pre- and post-breeding season and FGMs were lower (p<0.05) in the post-breeding season compared to pre-breeding season. Males fed with carcass had lower (p<0.001) FGM than males that were not fed carcass during both the pre-breeding and the

**Funding:** Funding was provided by The Davee Foundation.

**Competing interests:** The authors have declared that no competing interests exist.

breeding season but not during the post-breeding season (p>0.05). Males supplemented with vitamin E had higher (p<0.001) FAM than non-supplemented males during the breeding season only. For both groups, FAM was highest (p<0.05) during the breeding season. In conclusion, adding carcass to the diet can reduce glucocorticoid production and adding vitamin E can increase testosterone during the breeding season, which may influence reproductive success.

## Introduction

Nutrition plays an important role in animal welfare as it can affect both physical and mental health. Proper nutrition for some species, like grazers, may be easier to replicate *ex situ*, but for other species, such as specialist carnivores, it may be more difficult to meet specific dietary needs [1,2]. Although it can be challenging to maintain healthy populations of specialized carnivores outside of their natural environment, for the black-footed ferret (ferret; *Mustela nigripes*) the *ex situ* population is the only reason why the species did not go extinct. Sylvatic plague (plague, *Yersinia pestis*), which is an introduced bacterial disease spread by fleas, along with habitat loss and the reduction of its natural prey, the prairie dog (*Cynomys* spp.), has resulted in the ferret's demise [3,4]. In the mid-1980s, the remaining ferret population was removed from the wild to initiate an *ex situ* breeding program in an effort to prevent extinction. Thirty-four years of successful management has led to the production of more than 9,600 black-footed ferrets at six facilities led by the United States Fish and Wildlife Service (USFWS) through a Species Survival Plan® (SSP; managed through AZA) and includes USFWS's National Black-footed Ferret Conservation Center (Colorado), Smithsonian Conservation Biology Institute (Virginia), Louisville Zoological Garden (Kentucky), Cheyenne Mountain Zoo (Colorado), Phoenix Zoo (Arizona), and Toronto Zoo (Ontario, Canada). Of those ferrets born, more than 4,400 have been released into 30 reintroduction sites across North America's Great Plains in the past 29 years [5]. However, the population is now facing reproductive challenges. Since 2000, the percentage of normal spermatozoa has declined from ~50% to 25%, and pregnancy success has decreased from 60% to 36% [6,7]. It is presumed that because there are no novel genes to bring into the population through natural means that this decline is related to inbreeding depression. Additionally, higher inbreeding coefficient (*F*) in ferrets correlated with lower ejaculate volume, sperm forward progression, and normal sperm [7]. Santymire and coauthors [7] also found a positive relationship between litter size and both normal sperm acrosome and sperm motility, demonstrating that having normal sperm is important for producing larger litters. There is evidence, however, that wild-born ferrets have improved semen quality and larger testes [8,9]. The latter results suggest that environmental conditions, rather than genetics, may be limiting ferret reproductive success.

Another factor that may affect ferret reproductive success is diet. In 2001, the Black-footed Ferret SSP switched from a diet consisting of 60% mink pellets and 40% rabbit meat (60/40 diet), which was based on domestic ferret (*Mustela putorius furo*) production diet, to a commercially available diet, Toronto Small Carnivore diet (TOR), which was produced by Milliken Meat Products Ltd. (Markham, Ontario, Canada). TOR is a soft diet formulated for carnivores that is composed of horsemeat supplemented with vitamins and minerals. It contains no bones, cartilage, organs, skin or connective tissue (millikenmeat.com/products) [10,11]. A previous study evaluated the impact of various diets including supplementing the TOR diet with whole carcass (hamster or prairie dog) and/or vitamin E on seminal traits in 55 male ferrets

[12]. Results indicated that TOR had excessively high concentrations of vitamin A (~43,000 IU/kg), which can compete with the absorption of vitamin E [13–16]. Specifically, vitamin E may protect testosterone production in the Leydig cells by reducing the incidence of lipid per-oxidation and/or increasing the anterior pituitary hormones, luteinizing hormone, and follicle stimulating hormone as observed in rats supplemented with the antioxidant [17]. Testosterone is important for the production of spermatozoa, thus, reproductive success. Without supple-mentation of vitamin E and/or carcass, ferrets on the TOR diet had lower sperm concentra-tions, testes volume, and sperm motility indices [12].

Supplementing carnivores' diet with whole prey or carcasses can also stimulate natural behaviors [18–20], and may reduce stereotypies [21,22] and be more enriching [20,23]. Incor-porating dietary enrichment items like live prey, carcass, bones, frozen fish or spices [24] can reduce the effect of stressors by lowering aggression, encouraging positive social interactions, and promoting natural food acquisition [20]. The reduction of stressors in the *ex situ* environ-ment is important for maintaining animal health. In response to long-term or repeated stress-ors, the hypothalamus-pituitary-adrenal (HPA) axis will produce glucocorticoids (GCs), which are steroidal hormones such as cortisol and corticosterone, to help the body return to homeostasis through energy mobilization [25]. However, chronic production of GCs can lead to suppression of normal behaviors, the immune system, and reproduction [26]. Therefore, monitoring GC production can provide some insight on how the environment is impacting animals. Because analyzing GCs via blood sampling can be stressful, non-invasive methods, such fecal glucocorticoid metabolite (FGM) analysis, are often used to monitor changes in stress physiology in wildlife [27]. FGM analysis is an important tool for monitoring the stress-ors of the *ex situ* environment especially for species that may be particularly sensitive [28].

In this study, we examined stress and reproductive biomarkers to determine the effects of vitamin E and whole carcass supplementation on fecal glucocorticoid (FGM) and androgen metabolite (FAM) production in ferrets. Our hypotheses were: 1) whole prey items added to the diet would lower FGM, and 2) because of the anti-oxidative properties of vitamin E, FGM would be lower and FAM would increase in ferrets fed a diet supplemented with vitamin E.

## Methods

This research was reviewed and approved by Lincoln Park Zoo's Research Committee (pro-posal #2007–005) and United States Fish and Wildlife Service (Carr, CO). No animals were anesthetized nor euthanized during this study. All animal studies conformed to the Guide for Care and Use of Laboratory Animals.

### Animals and dietary treatments

Ferrets ($n$ = 42 males; ~1 kg; 1 to 3 years old) were housed individually at the National Black-footed Ferret Conservation Center (FCC; Carr, CO) in (1.0 m x 1.3 m x1.0 m) indoor cages. Lighting was both natural (provided by skylights) and artificial (via fluorescent illumination; set to natural photoperiod). Ferrets were fed 75–100 g of TOR, which was handled according to manufacturer's recommendations, daily and were provided water *ad libitum*. Males were assigned randomly to one of the following four diet treatments, including 16 juveniles (one year olds) and 26 adults (two and three year olds, which is the prime breeding age [29]),: 1) horsemeat diet with no supplementation (control); 2) horsemeat diet + vitamin E (D-α-tocopherol; 400 IU/kg diet dry matter basis (DMB); Stuart Products, Inc., Bedford, TX); 3) horsemeat diet + vitamin E (400 IU/kg diet DMB) + carcass item (two hamsters or prairie dogs pieces weighing approximately 75 to 100 g per week); 4) horsemeat diet + carcass item (two hamsters or prairie dogs pieces weighing approximately 75 to 100 g per week). The

hamsters were from a colony that was raised at FCC. The prairie dogs came from a partnership between the U.S. Fish and Wildlife Service and local county government agencies in Colorado who removed prairie dogs in accordance with their land management plans. Complete dietary analyses of TOR, hamster and prairie dog carcasses are described in Santymire et al. [12]. A subset of samples from adult males ($n$ = 15) was used to evaluate the effects of supplementation of vitamin E and/or carcass on FAM, even though previous research demonstrated that age (1 to 5 years of age) did not affect semen quality and testosterone concentrations [30].

### Sample collection and processing

Fecal samples were collected in the morning three times a week from November and December (pre-breeding season), February through May (breeding season) and June (post-breeding season). The diet change occurred in January and continued through June for all treatment groups. All samples were stored at -20˚C until processing. Fecal samples were shipped to and processed at Lincoln Park Zoo's endocrinology laboratory (Chicago, IL, USA). Feces were processed using previously described methods [31]. Briefly, samples were dried on a lyophilizer (Thermo Fisher Scientific, Inc., Waltham, MA) and were then crushed using a rubber mallet. The powdered samples were weighed (0.02 ± 0.002g) and 0.5 ml of 90% ethanol: distilled water was added to each aliquot. Samples were mixed (Glas-Col, Terre Haute, IN) with the ethanol for 30 mins and centrifuged (1,500 rpm; 20 minutes). The supernatant was then poured into clean test tubes. Residual feces were resuspended in another 0.5 ml of 90% ethanol, vortexed for 30 seconds, and centrifuged for 15 minutes. The supernatant was added to the first extract and was evaporated with air and heat (60˚C). The extracts were reconstituted in 0.5 ml of phosphate-buffered saline (0.2M $NaH_2PO_4$, 0.2Ms $Na_2HPO_4$, NaCl) before analysis.

### Hormone metabolite analysis

We analyzed the samples for FGM via a corticosterone enzyme immunoassay (EIA) that was previously validated for ferrets [31]. Briefly, the corticosterone antiserum (CJM006; provided by C. Munro, University of California, Davis, CA) and horseradish peroxidase (HRP; provided by C. Munro) were used at dilutions of 1:6,000 and 1:20,000, respectively. Antiserum cross-reactivities for corticosterone were previously described [32]. Biochemical validations of parallelism between binding inhibition curves of fecal extract dilutions and the corticosterone standard and significant recovery (>90%) of exogenous corticosterone (1.95–1,000 pg/50 μl) has been previously published [31]. Assay sensitivity was 1.95 pg/50 μl and intra- and inter-assay coefficients of variation were <10%.

FAMs were analyzed using a testosterone EIA. The polyclonal antiserum (R156/7) and HRP were provided by C. Munro and used at dilutions of 1:10,000 and 1:30,000, respectively. Antiserum cross-reactivities for testosterone were previously described [32]. The testosterone EIA was validated for ferrets by demonstrating: 1) parallelism between binding inhibition curves of fecal extract dilutions (1:160–1:10,240) and the testosterone standard ($R^2$ = 0.994); and 2) significant recovery (> 90%) of exogenous testosterone (1.17–300 pg/50 μl) added to fecal extracts (1:10,000; $y$ = 1.48$x$ − 2.00, $R^2$ = 0.997). Assay sensitivity was 2.3 pg/50 μl and intra- and inter-assay coefficients of variation were <10%.

### Data analysis

We first used a linear mixed effects model to identify the important fixed effect(s) among carcass treatment, vitamin E treatment, age class (juvenile or adult; FGM only), and all possible interactions between these fixed effects on each hormone. We included individual males nested within months nested within seasons (pre-, during, and post-breeding season) as

**Table 1. Model parameters from the fitted linear mixed effects model for FGM.**

| Fixed effects | Value | Standard Error | DF | t-value | p-value |
|---|---|---|---|---|---|
| (Intercept) | 8.06 | 0.10 | 3499 | 77.73 | 0.00 |
| Carcass treatment | 0.15 | 0.07 | 280 | 2.04 | 0.04 |
| Vitamin E treatment | 0.03 | 0.07 | 280 | 0.45 | 0.65 |
| Age class | 0.08 | 0.08 | 280 | 0.97 | 0.33 |
| Carcass * Vitamin E | 0.08 | 0.10 | 280 | 0.81 | 0.42 |
| Carcass * Age class | 0.03 | 0.12 | 280 | 0.26 | 0.80 |
| Vitamin E * Age class | -0.17 | 0.11 | 280 | -1.50 | 0.13 |
| Carcass * Vitamin E * Age class | -0.12 | 0.16 | 280 | -0.78 | 0.44 |
| Random effects | Standard Deviation | % Variance explained | | | |
| Season | 0.15 | 17.9 | | | |
| Month within season | < 0.001 | 0.73 | | | |
| Male within month within season | 0.31 | 36.4 | | | |
| Residual | 0.39 | 45.7 | | | |

random effects. We log-transformed the response variables (FAM and FGM) for analysis. We fit these models in R version 3.5.2 [33] using the package *nlme* [34]. Models were fit using the REML method and assuming no within-group correlations. We estimated effects of the random factors using variance components analysis and tested significance using likelihood ratio tests. We measured the goodness of fit of the models by examining the correlation between model predictions and the observed data. We then examined the effects over time (across seasons) for the treatment identified as important from these models using a Friedman test (a non-parametric repeated measures ANOVA) and the Student-Newman-Keuls method for multiple comparisons using Sigma Stat (version 11.0; Systat Software, Inc.; Chicago, IL). Finally, we examined the effects of the important treatments within each season, using a t-test if data were normal or Mann-Whitney U Rank Sum test otherwise. For all statistical analyses, we used the significance threshold of $p < 0.05$.

## Results

The mixed effects models showed no significant interactions among the three fixed effects for either FGM (Table 1) or FAM (Table 2). All three random effects were included in the models as the likelihood ratio tests indicated removing any of them would significantly reduce model fit ($p < 0.05$ in all cases). The models showed good fit to the dataset, with $R^2$ values for the

**Table 2. Model parameters from the fitted linear mixed effects model for FAM.**

| Fixed effects | Value | Standard Error | DF | t-value | p-value |
|---|---|---|---|---|---|
| (Intercept) | 8.94 | 0.37 | 643 | 24.01 | 0.00 |
| Carcass treatment | -0.02 | 0.14 | 95 | -0.12 | 0.90 |
| Vitamin E treatment | 0.25 | 0.14 | 95 | 1.85 | 0.07 |
| Carcass * Vitamin E | -0.22 | 0.20 | 95 | -1.08 | 0.28 |
| Random effects | Standard Deviation | % Variance explained | | | |
| Season | 0.60 | 32.3 | | | |
| Month within season | 0.22 | 12.0 | | | |
| Male within month within season | 0.46 | 24.9 | | | |
| Residual | 0.57 | 30.9 | | | |

**Table 3. Goodness of fit statistics for the two models.**

|  | AIC | BIC | logLikelihood | R^2 | t-value | DF | p-value |
|---|---|---|---|---|---|---|---|
| FGM | 4406.71 | 4481.58 | -2191.36 | 0.68 | 57.48 | 3791 | < 2.2e-16 |
| FAM | 1503.53 | 1540.43 | -743.77 | 0.77 | 33.47 | 746 | < 2.2e-16 |

correlation between observed and predicted values being 0.68 and 0.77 for FGM and FAM, respectively (Table 3).

For FGM, there was a significant effect of carcass treatment (p<0.05) but no effect of vitamin E or age class (Table 1). Variation between different measurements for each male within a month explained 36.4% of the variation not explained by the fixed effects, and variation between seasons explained 17.9% of the variation (Table 1). Because there were no interactions (p = 0.42) between the carcass and vitamin E treatments, the four treatment groups were pooled into two groups of carcass and no carcass. In the pre-breeding season before supplementation began, males in the carcass treatment had lower (U = 129,424.0, p = 0.004) FGM than non-supplemented males even though individuals were randomly selected for each treatment. Once the carcass was fed during the breeding season, FGMs declined (p<0.001) in males who received carcass; however, males that did not receive carcass had higher (p<0.05) FGMs (Fig 1). Comparing the treatments, FGM was lower (U = 517,553.5, p<0.001) in the carcass fed males in than males who were not fed carcass during the breeding season. In the post-breeding season, FGM were comparable (U = 15,245.0, p = 0.67) between treatments; however, while FGM for carcass fed males was similar (q = 0.951, p>0.05) between during and post-breeding season periods, non-supplemented males had lower FGMs in the post-breeding season compared to pre- (q = 11.118, p<0.05) and during (q = 10.734, p<0.05) the breeding season (Fig 1).

For FAM, there was an effect of vitamin E at the level of p = 0.07, and no effect of carcass treatment (p>0.05; Table 2). Variation between measurements within a month and between different seasons accounted for most of the unexplained variation (24.9% and 32.3%, respectively; Table 2). Because there were no interactions (p = 0.28) between the carcass and vitamin E treatments, the four treatment groups were pooled into two groups of vitamin E and no vitamin E supplementation. In the pre-breeding season before supplementation began, males in the vitamin E treatment had lower (U = 6594.0, p = 0.026) FAM than non-supplemented males. When supplementation began in the breeding season, males receiving vitamin E had higher ($t_{557}$ = -3.91, p<0.001) FAMs than males that were not supplemented, while both treatment groups had increases in FAM (vitamin E, q = 2.907, p<0.001; no vitamin E, q = 4.289, p<0.05) compared to the pre-breeding season (Fig 2). In the post-breeding season, supplementation of vitamin E had no effect (U = 641.0, p = 0.127) on FAM, but FAM was the lowest (compared to pre-breeding:q = 6.261, p<0.05; during: q = 6.483, p<0.05) for the vitamin E group during this time, while males that did not receive vitamin E had similar FAM as pre-breeding (q = 2.236, p>0.05), but lower than the breeding season (q = 4.585, p<0.05; Fig 2).

## Discussion

Our goal was to determine the effects of supplementing whole carcass and vitamin E on ferret stress and reproductive physiology. Our previous research examined the relationship between vitamin E and carcass supplementation on ferret seminal characteristics, and determined that the addition of the carcass twice a week improved semen traits and testes volume [12]. Here we specifically wanted to examine the effects on hormonal production. In general, lower psychological stress could reduce HPA axis activity, promote health, and increase reproduction

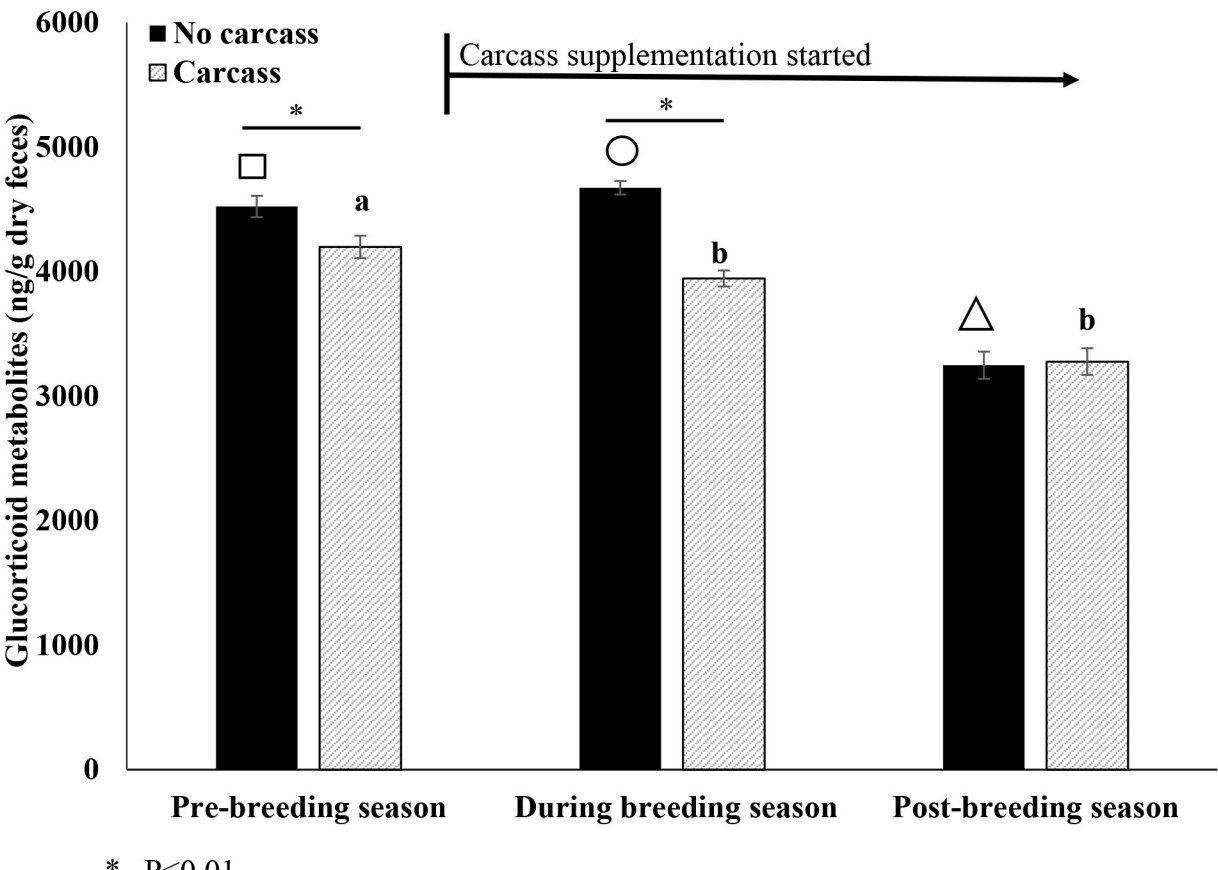

**Fig 1. Mean (± SEM) fecal glucocorticoid metabolites (FGM) across season (pre-, during and post-breeding season) and treatment (carcass vs. no carcass).** Different superscripts (a, b, c) depict differences (P < 0.05) across time in ferrets fed carcass. Different symbols (square, triangle, circle) depict differences (P < 0.05) across time in non-supplemented ferrets. Finally asterisks indicate differences (P < 0.05) between treatments within each time period.

[35,36]. Overall, we found an effect of carcass feeding but not vitamin E on FGMs. For FAMs, there was an effect of vitamin E supplementation but no effect of feeding carcass.

For FGMs, we found ferrets that were fed carcass twice weekly had lower FGM during the breeding season compared to the pre-breeding season. Interestingly, males that were not fed carcass had increased FGM during the breeding season. Initially before the supplementation began, the non-carcass treatment group had significantly higher FGMs (~8% higher). This is was an artifact of random selection of individuals for each treatment. However, upon carcass supplementation, the difference between the treatment groups increased (~18% higher) during the breeding season. This increase in difference could be a result of increased time spent foraging and a reduction of abnormal behaviors; however, we were unable to measure behavior to determine exactly how the carcass directly affected time spent foraging. Previously, environmental enrichment reduced FGM in ferrets, but this finding was sex and age dependent with juvenile males demonstrating greater decline in FGM compared to adult males and females [31]. Here, we only used males in the study and did not observe a difference in response between juveniles and adults. As all ferrets had lower FGM during the post-breeding season, FGM may be driven by the breeding season. However, the carcass supplementation did lower FGM enough during breeding season that was not significantly different than post-breeding season.

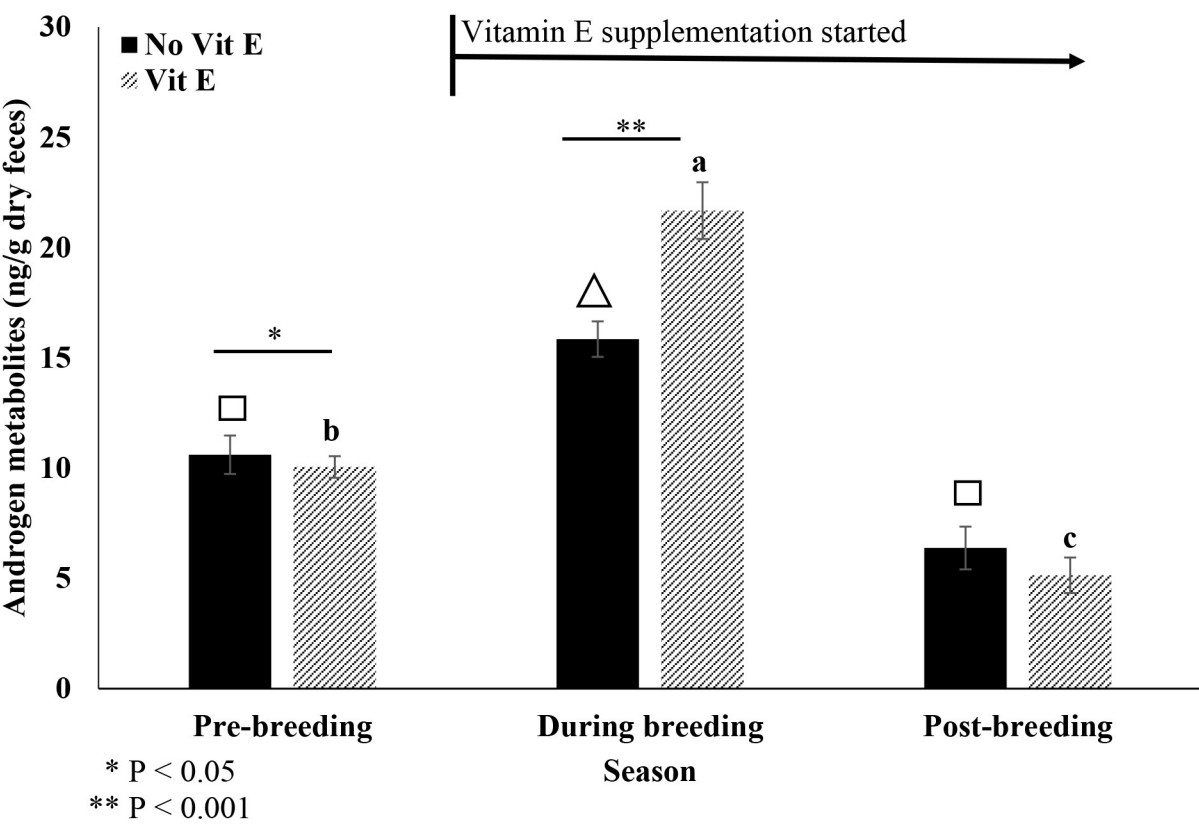

**Fig 2. Mean (± SEM) fecal androgen metabolites (FAM) across season (pre-, during and post-breeding season) and treatment (vitamin E vs. no vitamin E).** Different superscripts (a, b, c) depict differences (P < 0.05) across time in ferrets supplemented with vitamin E. Different symbols (square, triangle) depict differences (P < 0.01) across time in non-supplemented ferrets. Finally, asterisks indicate differences (P < 0.05) between treatments within each time period.

Naturally, the breeding season may be more stressful for the ferret as this is the time that wild males would set up territories and compete with conspecific males to gain access to females coming into estrus [37,38]. Although physical competition does not occur *ex situ*, males are housed in the same room as other males and females; therefore, there may be perceived conflict with other males especially when females are coming into estrus. Poessel and colleagues [31] also observed an increase in ferret FGM as animals entered the breeding season. Romero [39] reviewed the literature and found that most species, especially amphibians, reptiles and birds, have higher baseline glucocorticoids during the breeding season; in general, mammals did not strictly follow this pattern.

Whole carcass-supplementation stimulated natural foraging behavior and maintained dental health in other carnivores, such as African lions (*Panthera leo*) [35], fishing cats (*Felis viverrina*) [40] and Sumatran tigers (*Panthera tigris sumatrae*) [35]. Furthermore, based on ferret museum specimens, individuals fed TOR had a higher prevalence of calculus and periodontal disease than counterparts fed the 60/40 diet; both had poorer dental health than wild ferrets [41]. Anecdotally, by adding carcass twice a week to the diet, the incidence of tartar and gingivitis in ferrets was reduced [42]. Improving the ferret's oral health may explain why FGMs were lower for carcass-fed males during the breeding season compared to the non-supplemented ferrets. Previous studies have indicated a correlation between HPA activation and periodontal disease (reviewed by [43]). Glucocorticoid-mediated suppression of the immune system, specifically T-cell inhibition, increases susceptibility to periodontal disease [44]. In the

rat, inflammatory responses to gingivitis resulted in increased HPA activation and higher circulating corticosterone concentrations [45]. Future studies should incorporate other biomarkers of stress, such as measuring oxidative stress, which is a result of an imbalance of antioxidants and pro-oxidants like reactive oxygen species (ROS) [46]. ROSs damage tissues and cells and can lead to infertility by affecting spermatozoa structure and function [47–50]. However, we did not observe a reduction in FGMs with the addition of vitamin E, which is an antioxidant that helps to minimize the impact of ROS [47]; therefore, the change in FGMs may be related to something other than inflammation.

In a previous study, male ferrets fed carcass were significantly heavier than males that were not supplemented [12]. This increase in weight did not affect overall body condition or reproductive success [12]; therefore, it is difficult to state whether it was a positive or negative change. However, the change in FGMs, observed here, could be a reflection of shifts in the ferrets' metabolic rate. Glucocorticoids, as the name suggests, are hormones that stimulate the production of readily available glucose to support the body in overcoming challenges whether they are psychological or physiological [26]. The increased FGMs for males that did not receive carcass supplementation may signal a shift in metabolic rate to maintain energy balance during the breeding season. Future research should include measuring thyroid hormones (TH), including thyroxine (T4) and triiodothyronine (T3), which are important biomarkers of stress and energy balance. THs regulate many metabolic and ontogenetic pathways and there is communication between HPA and the H-P-Thyroid (HPT) axes [reviewed by 51]. Specifically for mammalian species, an acute HPA stimulate may result in lower TH production, which can affect metabolic rates [51]. Therefore, by analyzing both THs and FGMs we can better understand ferret homeostasis [52].

As observed previously in the ferret and in closely related species including the domestic ferrets (*Mustela putorius furo*) and Siberian polecats (*M. eversmanni*) [53], FAM values peaked during the breeding season for all treatments. However, we observed higher FAMs in males supplemented with vitamin E compared to the non-supplemented males during the breeding season. Vitamin E is known to play a role in steroidal hormone production and even in activating spermatogenesis [17,54]. Specifically, vitamin E stimulates the production of pituitary gonadotropins increasing plasma testosterone [17]. Vitamin E supplementation on rat Leydig cells resulted in increased testosterone production [55]. And vitamin E also reduces lipid peroxidation in the semen and improves sperm motility in men [56], stallions [57], sheep [58] and boars [59]. Our previous research examined the relationship between vitamin E and carcass supplementation on ferret seminal characteristics, showing that the addition of the carcass twice a week improved semen traits and testes volume [12]. Interestingly, in the current study we found no effect of carcass feeding and no interaction between vitamin E and carcass feeding on FAM production.

While we observed a decrease in FGMs during the breeding season when the ferret was supplemented with carcass and an increase in FAMs with supplemental vitamin E, a previous report found no effect on siring abilities [12]. Boosting reproductive success in the ferret recovery program is our primary conservation goal [60] as pregnancy success declined from 80% in 1990s to 36% across the SSP [61]. Normal spermatozoa in male ejaculate has declined from 50% to 25% [6]. Since demonstrating the importance of normal spermatozoa and sperm motility for pregnancy in the ferret [7], we have been investigating factors that could be attributing to ferret infertility. Declining reproductive rates may be related to inbreeding depression. The ferret population is a closed population with no opportunity to introduce novel genes to the species, at least not naturally. However, we have observed an improvement in semen quality in wild-born males [8,9]. Specifically, wild-born ferrets have nearly double the percent of normal spermatozoa in the ejaculate, while *ex situ*-born males living in the wild for at least one year

still have a comparable quality to the *ex situ* population [62]. Therefore, we surmise that an environmental condition, such as diet, housing, lighting and/or other husbandry factors, may be affecting ferret physiology and reducing reproductive success. These effects may be altered by being born into wild living conditions, but they cannot be reversed in *ex situ*-born individuals. Because diet may be the variable that is leading to infertility in the ferret population, our next step is to conduct a multi-year study across several generations of the *ex situ* population to test whether we observe similar trends noted in wild populations.

Another important finding is that variation in FGMs between multiple measurements within a month for each male explained 36.4% of the variation not explained by the fixed effects. For FAM analysis, variation among measurements within a month and among different seasons accounted for relatively high levels of the unexplained variation (32.3% and 24.9%, respectively). This variability over time demonstrates the importance of collecting multiple, frequent samples from individuals when using fecal hormone metabolite analysis particularly for understanding stress physiology.

## Conclusions

In conclusion, this study is the first to examine the effects of vitamin E and carcass supplementation on hormone concentrations in the ferret. While FAMs increased in all males, ferrets supplemented with vitamin E had significantly higher FAMs than the non-supplemented males during the breeding season. Furthermore, the addition of carcass to the ferret diet resulted in lower FGMs during the breeding season compared to males not fed carcass, which had increased FGMs compared to pre-breeding season values. The reduction in FGMs for the carcass-fed males may be attributed to improved dental health and/or the enriching nature of feeding on the carcass; however, this effect did not continue in the post-breeding season. This finding suggests that FGMs are affected by breeding season but we cannot distinguish between changes in FGM due to metabolic changes and/or psychological stress. Further research is needed to determine the effects of carcass feeding on ferret metabolism and stress physiology by examining other biomarkers, such as THs. Neither an increase in FAMs nor a decrease in FGMs during the breeding season influenced reproductive success [12]. Our results justify the need to continue studying the effects of diet on reproductive and stress physiology in the ferret. We suggest conducting a long-term study that extends over multiple years to determine the effects of hormonal changes on ferret reproductive biology.

## Supporting information

**S1 File. Fecal hormone metabolite data including age class, diet treatment and season.** (XLSX)

## Acknowledgments

We thank Diana Armstrong, Jenna Stewart, Christina Pawlik, for laboratory assistance. We also thank the staff at the National Black-footed Ferret Conservation Center for assistance with the project.

## Author Contributions

**Conceptualization:** Rachel M. Santymire, Shana R. Lavin, Heather Branvold-Faber, Julie Kreeger, Michelle Rafacz, Paul Marinari.

**Data curation:** Rachel M. Santymire.

**Formal analysis:** Judy Che-Castaldo.

**Funding acquisition:** Rachel M. Santymire.

**Investigation:** Rachel M. Santymire, Michelle Rafacz, Paul Marinari.

**Methodology:** Rachel M. Santymire, Shana R. Lavin, Heather Branvold-Faber, Julie Kreeger, Michelle Rafacz.

**Project administration:** Rachel M. Santymire, Heather Branvold-Faber, Julie Kreeger, Paul Marinari.

**Supervision:** Rachel M. Santymire, Heather Branvold-Faber, Paul Marinari.

**Validation:** Rachel M. Santymire, Julie Kreeger.

**Visualization:** Rachel M. Santymire, Judy Che-Castaldo.

**Writing – original draft:** Rachel M. Santymire, Judy Che-Castaldo.

**Writing – review & editing:** Rachel M. Santymire, Shana R. Lavin, Heather Branvold-Faber, Julie Kreeger, Judy Che-Castaldo, Michelle Rafacz, Paul Marinari.

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
