## [Decision Letter · Decision Letter 0]

28 May 2020

PONE-D-20-09363

Diet supplementation reduced fecal glucocorticoid metabolites and increased androgen metabolites in male black-footed ferrets (Mustela nigripes)

PLOS ONE

Dear Dr. Santymire,

Thank you for submitting your manuscript to PLOS ONE. After careful consideration, we feel that it has merit but does not fully meet PLOS ONE’s publication criteria as it currently stands. Therefore, we invite you to submit a revised version of the manuscript that addresses the points raised during the review process.

I completely agree with both the reviewers that the manuscript is required a substantial revision before any decision. Provide detailed regular diet given during the experiment.   

We look forward to receiving your revised manuscript.

Kind regards,

Govindhaswamy Umapathy, PhD

Academic Editor

PLOS ONE

Journal Requirements:

'All animal experiments conformed to the Guide for Care and Use of Laboratory Animals and were approved by the Lincoln Park Zoo Research Committee'.

(a)  Please state whether the provided ethics committee contains animal welfare experts or whether an animal ethics or IACUC committee reviewed and approved the study. Please provide the full name of the committee that reviewed and approved the study

(b) Once you have amended this/these statement(s) in the Methods section of the manuscript, please add the same text to the “Ethics Statement” field of the submission form (via “Edit Submission”).

3. At this time, we request that you  please report additional details in your Methods section regarding animal care, as per our editorial guidelines:

(a) Please state the source of the ferrets used in the study

(b) Please state whether the ferrets were euthanized at the end of the study

(c) Please describe the care received by the animals, including the frequency of monitoring and the criteria used to assess animal health and well-being.

(d) Please provide the source of the hamster and prairie dog carcass items used during the diet treatment study.

Thank you for your attention to these requests.

4. Please note that PLOS does not permit references to “data not shown.” Authors should provide the relevant data within the manuscript, the Supporting Information files, or in a public repository. If the data are not a core part of the research study being presented, we ask that authors remove any references to these data."

5. To comply with PLOS ONE submission guidelines, in your Methods section, please provide additional information regarding your statistical analyses, including the threshold set for statistical significance. For more information on PLOS ONE's expectations for statistical reporting, please see https://journals.plos.org/plosone/s/submission-guidelines.#loc-statistical-reporting.

"Funding was provided by The Davee Foundation."

Reviewers' comments:

Reviewer's Responses to Questions

**Comments to the Author**

1. Is the manuscript technically sound, and do the data support the conclusions?

Reviewer #1: Partly

Reviewer #2: No

2. Has the statistical analysis been performed appropriately and rigorously? 

Reviewer #1: Yes

Reviewer #2: Yes

3. Have the authors made all data underlying the findings in their manuscript fully available?

Reviewer #1: Yes

Reviewer #2: Yes

4. Is the manuscript presented in an intelligible fashion and written in standard English?

Reviewer #1: Yes

Reviewer #2: Yes

5. Review Comments to the Author

Reviewer #1: Abstract

Line 32: enzyme immunoassay (no “s”)

Line 39: Remove the extra “the”; explicitly state that age was not significant so the reader is not looking for those data in the lines that follow

Line 41: Does “supplemented” refer to vitamin E or carcass? The abstract needs to be more consistent throughout so that it is clear what “supplemented” is referring to in each sentence.

Introduction

Line 63: No need to repeat “muscle”

Line 63-64: The authors have already stated it is difficult to replicate.

Line 75: Suggest “improving physical health” (vs. dental alone)

Line 78: Do the authors mean “psychological” here (since physiological was discussed in the previous paragraph and the current paragraph seems to focus on behavior)?

Line 102: normal sperm acrosome

Line 106: Please quantify “nearly coincides”

Line 119-122: The authors did not really examine stress or psychological enrichment. They examined adrenal and reproductive biomarkers, and the first hypothesis should be revised to state simply that FGM would be lower in ferrets fed whole prey items.

Overall, this section is well-written, interesting, and presents good reason for conducting the study.

Methods

Lines 135-137: Perhaps this can be combined with the sentences at the start of this section to consolidate?

Line 138: Sample size was stated in the previous paragraph but with less detail regarding sample sizes of the age groups. Consolidate all this information in the same sentence.

Line 143: Presumably, “adult” is the 2 and 3 year-olds? The authors have not yet talked about the samples, so it might be clearer to state simply “a subset of adult males…was used to evaluate…” Lastly, there are 4 treatment groups, but this subset in which FAM will be measured appears to be divided into only 2, yet is to be used for evaluating effects of vit. E and/or carcass. If it is for evaluating vit. E plus either carcass or no carcass, perhaps in that case the authors can state, “a subset of adult males in groups 2 and 3 was used to evaluate…”. Otherwise, there would seem to be two missing groups from this subset (no vit E + carcass; no vit E no carcass)? Indeed, the results suggest the comparison of importance is between vitamin E versus no vitamin E, so is that really group 3 vs. group 4 were vitamin E supplementation differs but both receive carcass?

Line 149: Were all these samples collected at the same time of day or, if not, have previous publications already demonstrated that the circadian rhythm in glucocorticoid concentrations does not appear in fecal samples collected at different times of day (i.e., the excretory lag time is sufficiently long to “average out” the daily variation in FGM concentrations)? If not collected at the same time of day and if not previously demonstrated that FGM do not vary across the day, perhaps this accounts for some of the important but unexplained variation in the statistical models mentioned in the discussion?

Line 150: It is unfortunate there could not be a breeding season, no diet change combination.

Line 151: Samples were stored at minus 20C

Line 154: were dried

Line 158: It would be clearer to state the residual fecal material was resuspended rather than the “samples” to avoid confusion with the first supernatant that was decanted.

Line 161: 0.5 ml

Line 165: immunoassay

Line 167: Remove extra “(“

Line 171, 172; Line 180, 181: Please provide standard values in pg/ml (or similar units) or provide assay details of how many µl are added per well so the reader is able to do the necessary calculations if they wish to repeat the methods. Also provide assay sensitivity in pg/ml or similar.

Line 174: Sample size already described earlier in this section

Line 174-175: analyzed by or analyzed using

Line 187-188: It was described earlier that juveniles were not sampled for the FAM analysis, only adults, so presumably this statement is irrelevant?

Clarification is needed in this section with respect to the experimental design for the FAM analysis. Otherwise, only minor edits to clarify details are needed.

Results

Line 222: Suggest “from pre-breeding season to breeding season time periods”

Line 220-231: I think these data will be more easily understood by the audience if the authors take us through the “story” told by these data chronologically. Currently, it describes what happens to each group of males (carcass-supplemented or not) separately, then compares them within the seasons, and we have to go back and remember what the differential responses were across those seasons. It could be better to say, “in pre-breeding season, males that would be fed carcass in the coming months had X FGM concentrations compared to those that would not be supplemented. Once carcass was fed, the males receiving it experienced X whereas those not receiving it experienced Y, and these concentrations differed from each other. By post-breeding season…” I think this will better communicate the message about FGM differences to the audience than the way it is currently presented.

Line 232: Suggest stating p = 0.07 rather than p < 0.10

Line 232-243: Again, I suggest laying these data out chronologically to better communicate the differences. Also, from Table 2, it looks like carcass had no effect on FAM and did not interact with vitamin E, but Figure 2 suggests there was no carcass supplementation going on at all. I suspect Figure 2 should state that treatment was “(carcass + vitamin E vs. carcass with no vitamin E)”. I was unclear about the break-down of the FAM groups in the methods as well. In addition to clarifying this in the methods, it would be helpful to state explicitly in the results that the described results for vit. E on FAM occurred for those on carcass in the same manner as they occurred for those not on carcass (or were only assessed for those on carcass?).

Overall, the data are all here, but the way they are organized leaves the reader to try and piece together a complete picture from the components, which I fear could result in losing the point the authors will want to make. If these paragraphs could be re-written to better focus the reader on the critical differences at the critical points in time, the authors will have the readers “on board” by the time we get to the discussion. The figures are quite nice and allow the audience to interpret the data (except it is still unclear if the vitamin E data are for animals that were on carcass, not on carcass, or both).

Discussion

Line 264-282: These previous findings provide the initial rational for testing supplemental carcass feeding, but perhaps this could be more briefly summarized since it was already mentioned in the introduction as well? The study itself does not generate any dental data, yet the topic kicks-off and dominates the first 20 lines of the discussion.

Line 282-288: While interesting, this deals with skeletal changes as a result of space/activity, which has nothing to do with the present study.

Line 288-293: Again, while that might be interesting to consider, none of the data presented in this study speaks to morphological changes, so this musing seems unnecessary.

Line 294-297: It could be reasonable that whole carcass is more psychologically enriching, but no behavioral data were presented in this study. There is no pre-carcass behavioral data (or non-FGM physiological data that could be used as an indicator of stress) to support the notion that the ferrets were stressed to begin with, nor were any behavioral (or other) data recorded to support the notion that stress was reduced by carcass feeding. The authors acknowledge the absence of this data in subsequent lines and suggest instances in other studies where behavior was improved by carcass feeding, but the suggested conclusion in ferrets is entirely based on supposition.

Glucocorticoids are metabolic hormones, and while the psychological stress response can require a change in metabolism, metabolic change associated with breeding condition and associated with a change in diet/digestion (both of which are featured in this study) can occur in the absence of a change in psychological state. The authors go too far in concluding that their data suggest a reduction in stress when the only parameter measured is glucocorticoid metabolites. The authors have demonstrated a change in metabolism, and as it was in the direction opposite of those ferrets that did not receive carcass during the breeding season, the change does indeed appear to be a result of the carcass supplementation, but the effects of that supplementation are unknown at this point in terms of why it caused a change in metabolism and what it means for the ferrets. Instead of a change in psychological stress, couldn’t FGM have been reduced because there was more bone and fibrous tissue to attempt to digest and less protein and simple carbohydrate? It would be very helpful to have weight data on these different groups to see what effect the diet change had on their weight gain/loss. Perhaps this information is available in the ZIMS records and can be analyzed to provide some insight here? Another option to consider if the authors feel strongly that the change in FGM is indeed related to stress is to test another biomarker for stress. Markers of oxidative stress have been measured using fecal samples in other species—possibly try Arbor Assays K059?

Line 311-313: If improved oral health is tied to carcass supplementation, and carcass supplementation continued in the post-breeding season, then shouldn’t FGM continue to be lower in the carcass group than the non-carcass group in the post-breeding season? Instead, they are the same. This suggests that the effects of carcass supplementation on FGM concentrations were strongest or most relevant to the biology of the ferret during breeding season in particular. It would seem that a relationship to oral health should be fairly consistent across the year, no? The authors do bring up the interesting idea of glucocorticoid-mediated suppression of the immune system in subsequent lines. I wonder if perhaps there is literature on how the immune response of ferrets during the breeding season is up- or down-regulated normally and if that could then have any relationship to the differential FGM response of the ferrets in this study?

Line 323: What is the situation in the managed population? Are males and females paired individually according to genetics or is there some opportunity for competition/choice?

Line 336: Is it possible to add the no-effect-of-carcass data to the FAM graph? It is interesting that although FGM increased in the non-carcass group whereas FGM decreased in the carcass group, the carcass group did not manage to have higher FAM (since FGM might suppress FAM). Perhaps FAM were higher in the carcass group, it just was not statistically significant, especially since the authors noted an improvement in semen characteristics in a previous study of carcass supplementation?

Line 328-347: Within this paragraph, the connection between the fact that FAM increased more during the breeding season in the ferrets that received vitamin E supplementation than those that did not (line 335-336) and the mechanism for that relationship and beneficial effects related to semen (lines 343-347) gets lost by the intervening information about the lack of relationships in this study between carcass and FAM and between vitamin E and FGM. The important vitamin E-FAM relationship probably needs to be in its own paragraph, while the absent carcass-FAM and vitamin E-FGM relationships can be in a separate paragraph. Your topic sentences for the paragraph beginning on line 367 could be topic sentences for these paragraphs, especially since they are not related to the subsequent lines of the paragraph in which they currently appear (370-375).

Line 360-364: It sounds like the issue might be an organizational development issue. Have you considered estrogens in the TOR diet or in the water source?

Line 369-373: Perhaps all this variation that is important but unexplained by the fixed effects also suggests, along with your fecundity data, that some other fixed effect must be tested in order to solve this ferret fertility issue.

Line 379-380: Again, I really do not think the authors can point to what about carcass feeding contributed to the lower FGM in carcass-fed ferrets with this data set and with the discussion points that are presented. If attributed to improved dental health or enriching qualities, I would think those effects should have continued into the post-breeding season. The authors should consider what could be special about the physiology and metabolism of the ferrets during the breeding season in particular that could better explain this interesting and possibly important finding.

The discussion brings up a lot of interesting points, but the authors need to discuss the data in this study a bit more thoroughly. There seems to be something missing from how the authors are thinking about the change in FGM that could be better explored. Possibly more data needs to be gathered (weight, other biomarkers of stress, even behavioral notes from keepers), and at this time the most that can be said is that a change occurred but additional study is necessary to understand why and to understand its potential effects. The authors should be very careful about concluding that there was any change in psychological stress—it is possible but remains unsupported at this time. The part of the discussion related to FAM could be better organized to draw attention to the essential finding(s) and how that relates to previous studies but does not fully solve the ferret reproduction issue.

The study on the whole is interesting and important, but clarification is necessary in the experimental design, and a more focused interpretation of the data presented by this study so that it neither over-states nor under-states possible conclusions is needed. Alternative interpretations should be sought for the FGM data as those provided in the discussion do not seem to adequately explain the findings.

Reviewer #2: Introduction -

1. The intro is far too long with unnecessary information. The authors don't include any measures about dentition or dental health so it is not needed in the intro. There were also no measures of behavior; therefore, much of Lines 61 - 81 are simply extra words and could be shortened substantially without losing any integrity of the paper's intent.

2. There should be some information included regarding the use of FGM and the link with behavior and stress. There are other more common measures of stress so some lit review provided regarding this link would be useful to the reader. The words "psychologically enriching" is rather vague and would be very difficult to prove.

Materials and Methods

1. Line 133. What is meant by "commercially available"? How is the diet formulated? Formulated to what recommendations, I would assume feline?

2. If the paper is titled "Diet supplementation", additional information should be provided about the diet inclusive of actual dietary analyses, particularly for vitamin E. Why were there no analyses of diets provided or any information about the diet itself? Were all the lots of meat diet the same? There is substantial variation in the concentrations of vitamin E in diets and certainly in whole prey depending on the diet of the prey item. It is unclear how this was or was not controlled for in this study and is a major fault considering the title. There also was no measurement of vitamin E from the ferrets so without either, it is not possible to draw a substantial conclusion about vitamin E as you have no idea if the requirement was met in first place.

2. Are there recommendations for ferrets or are all diets formulated for cats?

3. Clarify how long the seasons lasted in days? The post breeding season was shorter than the other 2 by a month. Why were 2 months not included for that season? Is that an appropriate amount of time for supplementation?

3. Since you had an age class, why not block the study by age?

Discussion

1. What is the relevance of lines 272 - 320 to the paper? It's a lot of extra words that are not important to this manuscript as none of that was measured here. There is indication that dental health was improved but it wasn't measured. It should not be included unless data are included. This is an entire section of the paper that is not substantiated by any data included in the study presented and should be removed. It is a nice lit review for the value of whole prey but is not relevant.

2. Starting with line 321: How are FGM's related to "stress"? Rather than spend so much time on 272-320 that means nothing to the measures of the study, it seems that there could be value in this particular measure but it is hardly discussed. If FGM's are a reliable measure of stress then this is very important for animal managers and could definitely be used to enhance the diets of managed carnivores. It's not clear in current form how FGM's are related to stress and this could really enhance this paper. and is only one measure of Why were other measures of "stress" not considered such as inflammatory markers, cortisol, etc...?

3. It would be beneficial for the authors to discuss the 2 main measures (FGM and FAM) as indicators of reproductive success? There is some indication that one (FGM) is related to stress while FAM is related to testosterone? With this being a generally read publication, it would be valuable to provide some rationale as to these 2 measures. Why they were selected and how they are more directly related to reproductive success.

4. The "no carcass" pre and breeding season values look like the error bars overlap? Therefore, how are those significantly different?

Conclusions

1. Your conclusions are not substantiated with the data you collected. You only measured FGM and FAM and both were only weakly linked to stress and reproductive success. The authors simply restated the results that FGM reduced with whole prey and FAM increased with E then there were subjective assumptions made about enrichment, dental health, feeding ecology and visitor perceptions...none of which were measures and conclusions cannot be drawn from the 2 measures obtained.

General

1. Is there any information on these values (FAM/FGM) during these time frames (pre, breeding and post breeding) while not on any supplementation (what are typical)? That would really strengthen the data if one could look at it comparatively. I assume that's why there was no diet change in the post breeding but what is it normally and how do those values change seasonally as a whole?

2. The title is misleading as "diet supplementation". It might be better as "Influence of supplemental vitamin E and whole prey on..."

6. PLOS authors have the option to publish the peer review history of their article (what does this mean?). If published, this will include your full peer review and any attached files.

Reviewer #1: No

Reviewer #2: No

---

## [Author Response · Author response to Decision Letter 0]

4 Aug 2020

The authors want to thank the Reviewers and the Editor. We appreciate the thoroughness of the review. All of these comments that have made this manuscript stronger. Our responses are in bold below each comment.

Abstract

Line 32: enzyme immunoassay (no “s”)

Corrected

Line 39: Remove the extra “the”; explicitly state that age was not significant so the reader is not looking for those data in the lines that follow

Corrected and added a statement about age (Lines 44-45)

Line 41: Does “supplemented” refer to vitamin E or carcass? The abstract needs to be more consistent throughout so that it is clear what “supplemented” is referring to in each sentence.

We changed “supplemented” to “fed carcass” throughout abstract and left “supplemented” to refer to vitamin E supplementation only. 

Introduction

Line 63: No need to repeat “muscle”

Corrected

Line 63-64: The authors have already stated it is difficult to replicate.

We removed the phrase in the sentence.

Line 75: Suggest “improving physical health” (vs. dental alone)

Agreed and corrected

Line 78: Do the authors mean “psychological” here (since physiological was discussed in the previous paragraph and the current paragraph seems to focus on behavior)?

Agreed and corrected

Line 102: normal sperm acrosome

Corrected

Line 106: Please quantify “nearly coincides”

Corrected

Line 119-122: The authors did not really examine stress or psychological enrichment. They examined adrenal and reproductive biomarkers, and the first hypothesis should be revised to state simply that FGM would be lower in ferrets fed whole prey items.

Agreed and corrected

Overall, this section is well-written, interesting, and presents good reason for conducting the study.

Thank you

Methods

Lines 135-137: Perhaps this can be combined with the sentences at the start of this section to consolidate?

Corrected

Line 138: Sample size was stated in the previous paragraph but with less detail regarding sample sizes of the age groups. Consolidate all this information in the same sentence.

Corrected

Line 143: Presumably, “adult” is the 2 and 3 year-olds? The authors have not yet talked about the samples, so it might be clearer to state simply “a subset of adult males…was used to evaluate…” Lastly, there are 4 treatment groups, but this subset in which FAM will be measured appears to be divided into only 2, yet is to be used for evaluating effects of vit. E and/or carcass. If it is for evaluating vit. E plus either carcass or no carcass, perhaps in that case the authors can state, “a subset of adult males in groups 2 and 3 was used to evaluate…”. Otherwise, there would seem to be two missing groups from this subset (no vit E + carcass; no vit E no carcass)? Indeed, the results suggest the comparison of importance is between vitamin E versus no vitamin E, so is that really group 3 vs. group 4 were vitamin E supplementation differs but both receive carcass? 

The reviewer makes an excellent point. To clarify the difference between FGM and FAM groups, we removed the “A subset (n=15)” sentence because it is stated below in the Methods describing FAM analysis. We also removed the carcass vs no carcass because they are also part of the 4 diet treatments as we investigated the interactions of vitamin E and carcass supplementation. These changes will make it clearer to the reader.

Line 149: Were all these samples collected at the same time of day or, if not, have previous publications already demonstrated that the circadian rhythm in glucocorticoid concentrations does not appear in fecal samples collected at different times of day (i.e., the excretory lag time is sufficiently long to “average out” the daily variation in FGM concentrations)? If not collected at the same time of day and if not previously demonstrated that FGM do not vary across the day, perhaps this accounts for some of the important but unexplained variation in the statistical models mentioned in the discussion?

Yes, all of the samples were collected the same time of day. We have added a sentence to the methods to clarify the time of day (Line 166). 

Line 150: It is unfortunate there could not be a breeding season, no diet change combination.

Control males were included on this analysis (Group 1). Line 150 is describing the timing (breeding season/pre- or post-breeding season) of the diet change.

Line 151: Samples were stored at minus 20C

Corrected

Line 154: were dried

Corrected

Line 158: It would be clearer to state the residual fecal material was resuspended rather than the “samples” to avoid confusion with the first supernatant that was decanted.

Corrected

Line 161: 0.5 ml

Corrected

Line 165: immunoassay

Corrected

Line 167: Remove extra “(“

Corrected

Line 171, 172; Line 180, 181: Please provide standard values in pg/ml (or similar units) or provide assay details of how many µl are added per well so the reader is able to do the necessary calculations if they wish to repeat the methods. Also provide assay sensitivity in pg/ml or similar.

Corrected

Line 174: Sample size already described earlier in this section

Corrected

Line 174-175: analyzed by or analyzed using

Corrected

Line 187-188: It was described earlier that juveniles were not sampled for the FAM analysis, only adults, so presumably this statement is irrelevant?

We deleted that sentence.

Clarification is needed in this section with respect to the experimental design for the FAM analysis. Otherwise, only minor edits to clarify details are needed.

To clarify, the only difference between the FGM and FAM analyses was age. No juveniles were included on the FAM because of the confounding effect of it being their first breeding season.

Results

Line 222: Suggest “from pre-breeding season to breeding season time periods”

Corrected

Line 220-231: I think these data will be more easily understood by the audience if the authors take us through the “story” told by these data chronologically. Currently, it describes what happens to each group of males (carcass-supplemented or not) separately, then compares them within the seasons, and we have to go back and remember what the differential responses were across those seasons. It could be better to say, “in pre-breeding season, males that would be fed carcass in the coming months had X FGM concentrations compared to those that would not be supplemented. Once carcass was fed, the males receiving it experienced X whereas those not receiving it experienced Y, and these concentrations differed from each other. By post-breeding season…” I think this will better communicate the message about FGM differences to the audience than the way it is currently presented.

Thank you for the suggestion, we have reorganized the FGM paragraph using your suggestions.

Line 232: Suggest stating p = 0.07 rather than p < 0.10

Corrected

Line 232-243: Again, I suggest laying these data out chronologically to better communicate the differences. Also, from Table 2, it looks like carcass had no effect on FAM and did not interact with vitamin E, but Figure 2 suggests there was no carcass supplementation going on at all. I suspect Figure 2 should state that treatment was “(carcass + vitamin E vs. carcass with no vitamin E)”. I was unclear about the break-down of the FAM groups in the methods as well. In addition to clarifying this in the methods, it would be helpful to state explicitly in the results that the described results for vit. E on FAM occurred for those on carcass in the same manner as they occurred for those not on carcass (or were only assessed for those on carcass?).

For both FGM and FAM we have added a sentence that states that because there were no interactions between vit E and carcass supplementation, for each hormone we condensed the 4 treatment groups into 2: Carcass vs. no carcass for FGM and Vit E and no vit E for FAM. We also did reorganize the FAM results as suggested.

Overall, the data are all here, but the way they are organized leaves the reader to try and piece together a complete picture from the components, which I fear could result in losing the point the authors will want to make. If these paragraphs could be re-written to better focus the reader on the critical differences at the critical points in time, the authors will have the readers “on board” by the time we get to the discussion. The figures are quite nice and allow the audience to interpret the data (except it is still unclear if the vitamin E data are for animals that were on carcass, not on carcass, or both).

Thank you, hopefully now with your helpful suggestions we have clarified the analysis and results.

Discussion

Line 264-282: These previous findings provide the initial rational for testing supplemental carcass feeding, but perhaps this could be more briefly summarized since it was already mentioned in the introduction as well? The study itself does not generate any dental data, yet the topic kicks-off and dominates the first 20 lines of the discussion.

We have greatly reduced this text. Some of the diet information was put into the introduction. The rest was either deleted or used to support our findings.

Line 282-288: While interesting, this deals with skeletal changes as a result of space/activity, which has nothing to do with the present study.

This text has been deleted.

Line 288-293: Again, while that might be interesting to consider, none of the data presented in this study speaks to morphological changes, so this musing seems unnecessary.

This text has been deleted.

Line 294-297: It could be reasonable that whole carcass is more psychologically enriching, but no behavioral data were presented in this study. There is no pre-carcass behavioral data (or non-FGM physiological data that could be used as an indicator of stress) to support the notion that the ferrets were stressed to begin with, nor were any behavioral (or other) data recorded to support the notion that stress was reduced by carcass feeding. The authors acknowledge the absence of this data in subsequent lines and suggest instances in other studies where behavior was improved by carcass feeding, but the suggested conclusion in ferrets is entirely based on supposition.

Glucocorticoids are metabolic hormones, and while the psychological stress response can require a change in metabolism, metabolic change associated with breeding condition and associated with a change in diet/digestion (both of which are featured in this study) can occur in the absence of a change in psychological state. The authors go too far in concluding that their data suggest a reduction in stress when the only parameter measured is glucocorticoid metabolites. The authors have demonstrated a change in metabolism, and as it was in the direction opposite of those ferrets that did not receive carcass during the breeding season, the change does indeed appear to be a result of the carcass supplementation, but the effects of that supplementation are unknown at this point in terms of why it caused a change in metabolism and what it means for the ferrets. Instead of a change in psychological stress, couldn’t FGM have been reduced because there was more bone and fibrous tissue to attempt to digest and less protein and simple carbohydrate? It would be very helpful to have weight data on these different groups to see what effect the diet change had on their weight gain/loss. Perhaps this information is available in the ZIMS records and can be analyzed to provide some insight here? Another option to consider if the authors feel strongly that the change in FGM is indeed related to stress is to test another biomarker for stress. Markers of oxidative stress have been measured using fecal samples in other species—possibly try Arbor Assays K059?

Since we can’t state the exact effects of the whole carcass we have modified our suggestion of the effect on stress and added information about weight changes due to the dietary treatments, which was published in Santymire et al., 2015. We also added the suggestions of adding another biomarker of stress, like thyroid hormonal and oxidative stress analysis, and that glucocorticoid changes may be an indicator of metabolic changes and not psychology stress.

Line 311-313: If improved oral health is tied to carcass supplementation, and carcass supplementation continued in the post-breeding season, then shouldn’t FGM continue to be lower in the carcass group than the non-carcass group in the post-breeding season? Instead, they are the same. This suggests that the effects of carcass supplementation on FGM concentrations were strongest or most relevant to the biology of the ferret during breeding season in particular. It would seem that a relationship to oral health should be fairly consistent across the year, no? The authors do bring up the interesting idea of glucocorticoid-mediated suppression of the immune system in subsequent lines. I wonder if perhaps there is literature on how the immune response of ferrets during the breeding season is up- or down-regulated normally and if that could then have any relationship to the differential FGM response of the ferrets in this study?

There is no literature on seasonal immune response changes in ferrets; however, we do state that both glucocorticoid production may more closely related to breeding season. We also added information on how FAMs increase too and because testosterone is known to suppress the immune system, perhaps the carcass provides a “distraction” by increasing foraging time and oral health (Lines 403-407).

Line 323: What is the situation in the managed population? Are males and females paired individually according to genetics or is there some opportunity for competition/choice?

Males and females are kept in the same room. Individuals are paired based on genetics (using a pedigree analysis). There are no opportunities for physical competition or mate choice. To clarify the statement, we added “wild” before “males”. In captivity, males still may have this competitive nature since they are sharing the same room as females. We have also added a sentence with this information.

Line 336: Is it possible to add the no-effect-of-carcass data to the FAM graph? It is interesting that although FGM increased in the non-carcass group whereas FGM decreased in the carcass group, the carcass group did not manage to have higher FAM (since FGM might suppress FAM). Perhaps FAM were higher in the carcass group, it just was not statistically significant, especially since the authors noted an improvement in semen characteristics in a previous study of carcass supplementation?

Because our model found no interaction effects of vitamin E and carcass feeding and no effect of carcass, we grouped these data by vitamin E supplementation for analyses and for the figure. We believe that this is the most appropriate way of grouping these results.

Line 328-347: Within this paragraph, the connection between the fact that FAM increased more during the breeding season in the ferrets that received vitamin E supplementation than those that did not (line 335-336) and the mechanism for that relationship and beneficial effects related to semen (lines 343-347) gets lost by the intervening information about the lack of relationships in this study between carcass and FAM and between vitamin E and FGM. The important vitamin E-FAM relationship probably needs to be in its own paragraph, while the absent carcass-FAM and vitamin E-FGM relationships can be in a separate paragraph. Your topic sentences for the paragraph beginning on line 367 could be topic sentences for these paragraphs, especially since they are not related to the subsequent lines of the paragraph in which they currently appear (370-375).

We have reorganized these paragraphs to make this connection.

Line 360-364: It sounds like the issue might be an organizational development issue. Have you considered estrogens in the TOR diet or in the water source?

Yes, we are investigating phytoestrogens in the TOR because it is composed of horse meat and the horses may be fed legumes. Preliminary findings are showing increased phyto E2s in captive males compare to wild males and females and captive females. We aren’t ready to discuss this in this paper, though.

Line 369-373: Perhaps all this variation that is important but unexplained by the fixed effects also suggests, along with your fecundity data, that some other fixed effect must be tested in order to solve this ferret fertility issue.

Agreed. We are currently making these dietary changes over generations and are investigating gene expression and have some interesting findings that we will be publishing after another year of data collection.

Line 379-380: Again, I really do not think the authors can point to what about carcass feeding contributed to the lower FGM in carcass-fed ferrets with this data set and with the discussion points that are presented. If attributed to improved dental health or enriching qualities, I would think those effects should have continued into the post-breeding season. The authors should consider what could be special about the physiology and metabolism of the ferrets during the breeding season in particular that could better explain this interesting and possibly important finding.

We have pointed out that the lower FGM did not remain significant in the post-breeding season and suggest that FGM changes may be more related to metabolism. We added a sentence suggesting adding the analysis of other biomarkers of stress like oxidative stress thyroid hormones.

The discussion brings up a lot of interesting points, but the authors need to discuss the data in this study a bit more thoroughly. There seems to be something missing from how the authors are thinking about the change in FGM that could be better explored. Possibly more data needs to be gathered (weight, other biomarkers of stress, even behavioral notes from keepers), and at this time the most that can be said is that a change occurred but additional study is necessary to understand why and to understand its potential effects. The authors should be very careful about concluding that there was any change in psychological stress—it is possible but remains unsupported at this time. The part of the discussion related to FAM could be better organized to draw attention to the essential finding(s) and how that relates to previous studies but does not fully solve the ferret reproduction issue.

We have made these suggested changes and believe that the discussion focuses more on this study’s findings and that FGM changes may be a reflection of other factors such as metabolic rate vs. psychology stress. We also reorganized the FAM discussion to focus on this study’s findings.

The study on the whole is interesting and important, but clarification is necessary in the experimental design, and a more focused interpretation of the data presented by this study so that it neither over-states nor under-states possible conclusions is needed. Alternative interpretations should be sought for the FGM data as those provided in the discussion do not seem to adequately explain the findings.

Thank you and your suggestions have helped us improve the discussion.

Reviewer #2: Introduction -

1. The intro is far too long with unnecessary information. The authors don't include any measures about dentition or dental health so it is not needed in the intro. There were also no measures of behavior; therefore, much of Lines 61 - 81 are simply extra words and could be shortened substantially without losing any integrity of the paper's intent.

We have reduced these two paragraphs down to one. 

2. There should be some information included regarding the use of FGM and the link with behavior and stress. There are other more common measures of stress so some lit review provided regarding this link would be useful to the reader. The words "psychologically enriching" is rather vague and would be very difficult to prove.

We have removed “psychologically” enriching. And we added a paragraph about monitoring stress physiology of ex situ wildlife.

Materials and Methods

1. Line 133. What is meant by "commercially available"? How is the diet formulated? Formulated to what recommendations, I would assume feline?

TOR was defined in the introduction. We deleted “commercially available” since we state the company that makes it. We also added to the introduction that TOR was specifically formulated for the ferret.

2. If the paper is titled "Diet supplementation", additional information should be provided about the diet inclusive of actual dietary analyses, particularly for vitamin E. Why were there no analyses of diets provided or any information about the diet itself? Were all the lots of meat diet the same? There is substantial variation in the concentrations of vitamin E in diets and certainly in whole prey depending on the diet of the prey item. It is unclear how this was or was not controlled for in this study and is a major fault considering the title. There also was no measurement of vitamin E from the ferrets so without either, it is not possible to draw a substantial conclusion about vitamin E as you have no idea if the requirement was met in first place.

In the Santymire et al. 2015 where the effect of this diet study on seminal characteristics is described, the diet analysis is included. We added a statement referring to that paper for dietary analysis in the methods (Lines 159-160). 

2. Are there recommendations for ferrets or are all diets formulated for cats?

For the ferret, we follow the dietary recommendations for the mink, which is also used in the Santymire et al. 2015 paper as a guideline for the dietary recommendations.

3. Clarify how long the seasons lasted in days? The post breeding season was shorter than the other 2 by a month. Why were 2 months not included for that season? Is that an appropriate amount of time for supplementation?

We believe that by listing the months, the readers will understand how long each “season” lasted. It would be ideal to collect samples for longer in the post-breeding season, but we were unable to collect samples past June because of kit production which requires a lot more staff time. Additionally, because the ferret is experience reproductive issues and we were wondering if GC production was suppressing reproduction, investigating stress physiology in the post-breeding season would not be helping to answer this question. Therefore, we did not prioritize staff time for fecal samples collection for this study.

3. Since you had an age class, why not block the study by age?

We included age class in the model for FGM but did not find it to be a significant predictor of FGM levels. We only measured FAM in adults and therefore did not include age class in that analysis. 

Discussion

1. What is the relevance of lines 272 - 320 to the paper? It's a lot of extra words that are not important to this manuscript as none of that was measured here. There is indication that dental health was improved but it wasn't measured. It should not be included unless data are included. This is an entire section of the paper that is not substantiated by any data included in the study presented and should be removed. It is a nice lit review for the value of whole prey but is not relevant.

We have reduced this section and included our findings supporting them with the literature.

2. Starting with line 321: How are FGM's related to "stress"? Rather than spend so much time on 272-320 that means nothing to the measures of the study, it seems that there could be value in this particular measure but it is hardly discussed. If FGM's are a reliable measure of stress then this is very important for animal managers and could definitely be used to enhance the diets of managed carnivores. It's not clear in current form how FGM's are related to stress and this could really enhance this paper. and is only one measure of Why were other measures of "stress" not considered such as inflammatory markers, cortisol, etc...?

Based on Reviewer 1’s comments, we have added how changes in FGMs might be a reflection of metabolism and added information about body weight changes. We also suggested that changes in FGM may be more related to the season than just the carcass feeding alone. We also suggesting investigating other biomarkers of stress like oxidative stress.

3. It would be beneficial for the authors to discuss the 2 main measures (FGM and FAM) as indicators of reproductive success? There is some indication that one (FGM) is related to stress while FAM is related to testosterone? With this being a generally read publication, it would be valuable to provide some rationale as to these 2 measures. Why they were selected and how they are more directly related to reproductive success.

We have added statements about the effect of GCs and testosterone on reproduction in the introduction.

4. The "no carcass" pre and breeding season values look like the error bars overlap? Therefore, how are those significantly different?

The SEM bars do not overlap in the pre- and breeding seasons between the carcass and no carcass treatments. The asterisks do correctly indicate that there is a significant difference.

Conclusions

1. Your conclusions are not substantiated with the data you collected. You only measured FGM and FAM and both were only weakly linked to stress and reproductive success. The authors simply restated the results that FGM reduced with whole prey and FAM increased with E then there were subjective assumptions made about enrichment, dental health, feeding ecology and visitor perceptions...none of which were measures and conclusions cannot be drawn from the 2 measures obtained.

In the conclusions, we have made suggestions for other studies to help determine why FGMs were reduced using other biomarkers. We have emphasized that these hormonal changes did not affect reproduction in the ferret and therefore need additional research to determine the etiology of infertility in the ferret.

General

1. Is there any information on these values (FAM/FGM) during these time frames (pre, breeding and post breeding) while not on any supplementation (what are typical)? That would really strengthen the data if one could look at it comparatively. I assume that's why there was no diet change in the post breeding but what is it normally and how do those values change seasonally as a whole?

We did have a control group (no vitamin E and no carcass feeding) to compare to these treatments and found that there were no effects or interactions. This treatment group would have defined what is “normal”. 

2. The title is misleading as "diet supplementation". It might be better as "Influence of supplemental vitamin E and whole prey on..."

We have changed the title to: Influence of supplementation of vitamin E and carcass feeding on fecal glucocorticoid and androgen metabolites in male black-footed ferrets (Mustela nigripes)

Journal Requirements:

We have corrected the style to meet PLOS ONE’s requirements.

'All animal experiments conformed to the Guide for Care and Use of Laboratory Animals and were approved by the Lincoln Park Zoo Research Committee'.

(a) Please state whether the provided ethics committee contains animal welfare experts or whether an animal ethics or IACUC committee reviewed and approved the study. Please provide the full name of the committee that reviewed and approved the study.

We have included more information about the Lincoln Park Zoo’s Research Committee in the Methods section.

(b) Once you have amended this/these statement(s) in the Methods section of the manuscript, please add the same text to the “Ethics Statement” field of the submission form (via “Edit Submission”).

 3. At this time, we request that you please report additional details in your Methods section regarding animal care, as per our editorial guidelines:

(a) Please state the source of the ferrets used in the study

We added this info to the Methods section.

(b) Please state whether the ferrets were euthanized at the end of the study

We added the final disposition of the ferrets to the Methods.

(c) Please describe the care received by the animals, including the frequency of monitoring and the criteria used to assess animal health and well-being.

We have added more information about care the ferrets receive to the Methods.

(d) Please provide the source of the hamster and prairie dog carcass items used during the diet treatment study.

We have added this information to the Methods.

Thank you for your attention to these requests.

4. Please note that PLOS does not permit references to “data not shown.” Authors should provide the relevant data within the manuscript, the Supporting Information files, or in a public repository. If the data are not a core part of the research study being presented, we ask that authors remove any references to these data."

We have provided data to the supplementary files.

5. To comply with PLOS ONE submission guidelines, in your Methods section, please provide additional information regarding your statistical analyses, including the threshold set for statistical significance. For more information on PLOS ONE's expectations for statistical reporting, please see https://journals.plos.org/plosone/s/submission-guidelines.#loc-statistical-reporting.

 We added our threshold of significance to the methods. For the post-hoc testing using the Student-Newman-Keuls test, the threshold of significance is P<0.05, therefore, this is how we report it.

"Funding was provided by The Davee Foundation."

We removed the funding statement from the Acknowledgments section and modified the Funding statement. "The author(s) received no specific funding for this work."

---

## [Decision Letter · Decision Letter 1]

25 Aug 2020

PONE-D-20-09363R1

Influence of vitamin E and carcass feeding supplementation on fecal glucocorticoid and androgen metabolites in male black-footed ferrets (Mustela nigripes)

PLOS ONE

Dear Dr. Santymire,

Thank you for submitting your manuscript to PLOS ONE. After careful consideration, we feel that it has merit but does not fully meet PLOS ONE’s publication criteria as it currently stands. Therefore, we invite you to submit a revised version of the manuscript that addresses the points raised during the review process.

As suggested by Reviewer 1 the manuscript requires some edits in introduction for better flow and minor edits in discussion. 

We look forward to receiving your revised manuscript.

Kind regards,

Govindhaswamy Umapathy, PhD

Academic Editor

PLOS ONE

Reviewers' comments:

Reviewer's Responses to Questions

**Comments to the Author**

1. If the authors have adequately addressed your comments raised in a previous round of review and you feel that this manuscript is now acceptable for publication, you may indicate that here to bypass the “Comments to the Author” section, enter your conflict of interest statement in the “Confidential to Editor” section, and submit your "Accept" recommendation.

Reviewer #1: (No Response)

Reviewer #2: (No Response)

2. Is the manuscript technically sound, and do the data support the conclusions?

Reviewer #1: Yes

Reviewer #2: Yes

3. Has the statistical analysis been performed appropriately and rigorously? 

Reviewer #1: Yes

Reviewer #2: Yes

4. Have the authors made all data underlying the findings in their manuscript fully available?

Reviewer #1: Yes

Reviewer #2: Yes

5. Is the manuscript presented in an intelligible fashion and written in standard English?

Reviewer #1: Yes

Reviewer #2: Yes

6. Review Comments to the Author

Reviewer #1: Introduction

Line 112: So, the problems with reproduction, at least from the perspective of semen quality, pre-dated the diet change, i.e., were unrelated to the diet change? If so, while it is still very important to get the diet as good as it can be for numerous reasons, this timing suggests the specific change made in 2001 is not the culprit for reproductive decline (though perhaps it is not helping the situation either).

Line 115: Missing the “)” after “Canada”

Methods

Lines 137: Seems to be something missing from the second half of this sentence

Lines 148-152: Groups are much clearer now—thank you

Line 163-164: The way it is currently presented with the parenthetical notes, it still seems like the control group was not sampled February-June, but I see your reply in the reviewer notes. I suggest stating that samples were collected from all 4 groups at each time point but no diet changes were initiated in treatment groups until February.

Results

Much clearer—thank you.

Discussion

Line 301: Remove “is largely”

Line 301-302: Suggested revision for clarity: “However, the carcass supplementation did lower FGM enough during breeding season that is was not significantly different than post-breeding season.”

Line 363-364: Intro states 60% to 35%

Line 376: Again, because the decline in reproduction was noted a year before the diet change, stating that “diet may be the variable that is leading to infertility” could be a bit strong. It might not help, to be sure, but I agree a more detailed study is required.

Line 401: I believe the authors mean to say “because glucocorticoids can suppress the immune response” based on the rest of the sentence discussing carcass addition and its potential benefits.

This section is better organized, the take-home-messages more clearly stated, and the statements better supported than in my first review. Alternative explanations are appropriately noted. Just a thought: If vit E supplementation begins in pre-breeding season as males are preparing for increased spermatogenesis (v. starting after the fact during breeding season), perhaps there might be a better outcome for reproduction…

Reviewer #2: Thank you for the changes that were made. Overall, it reads much better. However, I am still struggling with the Introduction. There are few reasons a paper should have a 3 1/2 page introduction. The first 3 paragraphs do not outline the problem at all but instead discuss nutrition, diet formulation and dental health. The gist of the paper's background very nicely starts on Line 88 with a very nice presentation of the history and issues. That then follows very nicely with the need to evaluate vitamin E starting on line 122. It seems that right after Line 128 you could add a bit about the behavior impact/stress on reproduction (starting from Line 71) then introduce the concept of whole prey. That would allow your ending objective statement to flow much better. Regardless the first 2 paragraphs do not benefit your paper in any way and take away from the actual problem that you don't start talking about until Line Line 88. This is about trying to enhance the diets for the BFF program not formulate diets and discuss nutrition for all zoo managed carnivores.

Line 67: What is "processed"? Do you mean basically a TOR that you define later (line 115)? A definition of processed should be included...is that simply ground meat that is complete? I think you mean TOR but you should define it sooner if you are referring to it here.

Line 141 should read Animals and Diet since that section is not just about the animals or it could be Animals and Treatments

Line 150: Add "diet" between Kg and Dry Matter Basis

Line 150 - 152: It is unclear if treatments 3 and 4 have the same whole/prey composition. Is "two hamsters or prairie dogs

pieces/week (3) the same thing as carcass item twice weekly (4)? Was there any attempt made to control/standardize the weight of carcass items for these 2 treatments. As it's currently written it doesn't appear to be the same.

7. PLOS authors have the option to publish the peer review history of their article (what does this mean?). If published, this will include your full peer review and any attached files.

Reviewer #1: No

Reviewer #2: No

---

## [Author Response · Author response to Decision Letter 1]

5 Oct 2020

in bold are our responses.

Reviewer #1: Introduction

Line 112: So, the problems with reproduction, at least from the perspective of semen quality, pre-dated the diet change, i.e., were unrelated to the diet change? If so, while it is still very important to get the diet as good as it can be for numerous reasons, this timing suggests the specific change made in 2001 is not the culprit for reproductive decline (though perhaps it is not helping the situation either).

Yes, the males collected from in 2001 were for artificial insemination program because they weren’t siring on their own and were genetically valuable. This might have been due to poor semen quality. We have removed this sentence in the introduction.

Line 115: Missing the “)” after “Canada”

Corrected

Methods

Lines 137: Seems to be something missing from the second half of this sentence

Corrected

Lines 148-152: Groups are much clearer now—thank you

You are welcome. Thank you for the helpful comments.

Line 163-164: The way it is currently presented with the parenthetical notes, it still seems like the control group was not sampled February-June, but I see your reply in the reviewer notes. I suggest stating that samples were collected from all 4 groups at each time point but no diet changes were initiated in treatment groups until February.

We have clarified sample collection and when diet change occurred.

Results

Much clearer—thank you.

You are welcome. Thank you for the helpful comments.

Discussion

Line 301: Remove “is largely”

Corrected

Line 301-302: Suggested revision for clarity: “However, the carcass supplementation did lower FGM enough during breeding season that is was not significantly different than post-breeding season.”

Corrected

Line 363-364: Intro states 60% to 35%

That was since 2000. Originally, whelping rates were around 80%. We clarified the sentence to make that point clear.

Line 376: Again, because the decline in reproduction was noted a year before the diet change, stating that “diet may be the variable that is leading to infertility” could be a bit strong. It might not help, to be sure, but I agree a more detailed study is required.

Hopefully, with the clarification about the males 2001 being AI males because they were not siring on their own, demonstrates why we are investigating diet further. With this several generations diet study, we hope to show results that suggest more than correlation, but causation (if possible without laboratory experimentation).

Line 401: I believe the authors mean to say “because glucocorticoids can suppress the immune response” based on the rest of the sentence discussing carcass addition and its potential benefits.

We did mean “testosterone”. We stated this because GCs went down during the breeding season but testosterone increased with the addition of carcass. The increased testosterone did not seem to improve reproductive success. So our point was that high testosterone has its costs, but the carcass might help to mitigate them along with providing enrichment/a distraction. But, we removed it and kept the conclusions more general.

This section is better organized, the take-home-messages more clearly stated, and the statements better supported than in my first review. Alternative explanations are appropriately noted. Just a thought: If vit E supplementation begins in pre-breeding season as males are preparing for increased spermatogenesis (v. starting after the fact during breeding season), perhaps there might be a better outcome for reproduction…

Thanks! We hope our 3 year diet change study will address this thought/idea.

Reviewer #2: Thank you for the changes that were made. Overall, it reads much better. However, I am still struggling with the Introduction. There are few reasons a paper should have a 3 1/2 page introduction. The first 3 paragraphs do not outline the problem at all but instead discuss nutrition, diet formulation and dental health. The gist of the paper's background very nicely starts on Line 88 with a very nice presentation of the history and issues. That then follows very nicely with the need to evaluate vitamin E starting on line 122. It seems that right after Line 128 you could add a bit about the behavior impact/stress on reproduction (starting from Line 71) then introduce the concept of whole prey. That would allow your ending objective statement to flow much better. Regardless the first 2 paragraphs do not benefit your paper in any way and take away from the actual problem that you don't start talking about until Line Line 88. This is about trying to enhance the diets for the BFF program not formulate diets and discuss nutrition for all zoo managed carnivores.

We have greatly reduced the first three paragraphs in the introduction. We made the suggested organizational changes, but kept a couple of sentences, such as the lead into nutrition’s importance to welfare, difficulty of specialized carnivore diets and the how the addition of prey items may reduce stress. We wanted some of the literature to set up the difficulties of feeding BFF, which is a specialized carnivore, and why adding carcass could affect GCs (through behavioral modifications).

Line 67: What is "processed"? Do you mean basically a TOR that you define later (line 115)? A definition of processed should be included...is that simply ground meat that is complete? I think you mean TOR but you should define it sooner if you are referring to it here.

Based on your suggested intro re-organization, this sentence has been deleted.

Line 141 should read Animals and Diet since that section is not just about the animals or it could be Animals and Treatments

Corrected

Line 150: Add "diet" between Kg and Dry Matter Basis

Corrected

Line 150 - 152: It is unclear if treatments 3 and 4 have the same whole/prey composition. Is "two hamsters or prairie dogs

pieces/week (3) the same thing as carcass item twice weekly (4)? Was there any attempt made to control/standardize the weight of carcass items for these 2 treatments. As it's currently written it doesn't appear to be the same.

Yes, it was the same. And, yes, weight of the carcass item was standardized. We clarified this in the text.

---

## [Editor Report · Decision Letter 2]

8 Oct 2020

Influence of vitamin E and carcass feeding supplementation on fecal glucocorticoid and androgen metabolites in male black-footed ferrets (Mustela nigripes)

PONE-D-20-09363R2

Dear Dr. Santymire,

We’re pleased to inform you that your manuscript has been judged scientifically suitable for publication and will be formally accepted for publication once it meets all outstanding technical requirements.

Kind regards,

Govindhaswamy Umapathy, PhD

Academic Editor

PLOS ONE
---

## [Editor Report · Acceptance letter]

13 Oct 2020

PONE-D-20-09363R2 

Influence of vitamin E and carcass feeding supplementation on fecal glucocorticoid and androgen metabolites in male black-footed ferrets (*Mustela nigripes*) 

Dear Dr. Santymire:

I'm pleased to inform you that your manuscript has been deemed suitable for publication in PLOS ONE. Congratulations! Your manuscript is now with our production department. 

Kind regards, 

on behalf of

Dr. Govindhaswamy Umapathy 

Academic Editor

PLOS ONE